# Development and validation of interpretable machine learning models for triage patients admitted to the intensive care unit

**Zheng Liu ⓘ, Wenqi Shu, Hongyan Liu, Xuan Zhang, Wei Chong***

Department of Emergency, The First Hospital of China Medical University, Shenyang, China

* wchong@cmu.edu.cn

## Abstract

### Objectives

Developing and validating interpretable machine learning (ML) models for predicting whether triaged patients need to be admitted to the intensive care unit (ICU).

### Measures

The study analyzed 189,167 emergency patients from the Medical Information Mart for Intensive Care IV database, with the outcome being ICU admission. Three models were compared: Model 1 based on Emergency Severity Index (ESI), Model 2 on vital signs, and Model 3 on vital signs, demographic characteristics, medical history, and chief complaints. Nine ML algorithms were employed. The area under the receiver operating characteristic curve (AUC), F1 Score, Positive Predictive Value, Negative Predictive Value, Brier score, calibration curves, and decision curves analysis were used to evaluate the performance of the models. SHapley Additive exPlanations was used for explaining ML models.

### Results

The AUC of Model 3 was superior to that of Model 1 and Model 2. In Model 3, the top four algorithms with the highest AUC were Gradient Boosting (0.81), Logistic Regression (0.81), naive Bayes (0.80), and Random Forest (0.80). Upon further comparison of the four algorithms, Gradient Boosting was slightly superior to Random Forest and Logistic Regression, while naive Bayes performed the worst.

### Conclusions

This study developed an interpretable ML triage model using vital signs, demographics, medical history, and chief complaints, proving more effective than traditional models in predicting ICU admission. Interpretable ML aids clinical decisions during triage.

**Data availability statement:** Publicly available datasets were analyzed in this study. These data can be found at https://mimic.mit.edu/. The datasets generated during and/or analysed during the current study are available in the figshare repository, accessible at: https://doi.org/10.6084/m9.figshare.26402761.v1.

**Funding:** This research was supported by the Education Department of Liaoning Province, China. The funding was awarded to Zheng Liu under the grant number LJ232410159024. The funders had no role in study design, data collection and analysis, decision to publish, or preparation of the manuscript.

**Competing interests:** The authors have declared that no competing interests exist.

## Introduction

For the past 20 years, the number of emergency patients has increased every year [1,2], and the overcrowding of emergency rooms has become an important health problem worldwide [3–7]. Similarly, the proportion of critically ill patients has also increased [8]. These patients need immediate treatment in the intensive care unit (ICU), but overcrowding causes them to wait longer in the emergency department (ED) [9]. Overcrowding in emergency rooms and delayed ICU admission can lead to a series of adverse consequences, such as increased mortality [10–13]. Timely risk stratification and diversion of emergency patients, which requires the help of an early warning system, is an effective measure to decrease emergency room congestion and improve the survival rate of critically ill patients [14–17].

Currently, the commonly used international triage standards include the Australian Triage Scale, Canadian Triage and Acuity Scale, Manchester Triage System, Emergency Severity Index (ESI) five-level triage system, Korean Triage and Acuity Scale, and National Early Warning Score [18,19]. Due to the limited information obtained during triage, these warning models were primarily based on subjective judgments by medical staff or the vital signs of patients, and previous studies had already demonstrated their shortcomings in predictive performance [16,20,21]. With the development of structured electronic medical records (sEMR) and machine learning (ML), additional variables can be collected during triage and included in the prediction model without increasing the workload of medical staff [17,22]. At present, there are few studies on the use of a ML method to predict admission to the ICU from complex data based on triage information, and the "black box" nature of ML limits its application in medical decision support [23,24]. The interpretability of ML models has always been one of the hotspots and challenges in research. Especially in the medical field, there is a higher demand for interpretability of model decisions to ensure transparency and reliability in the decision-making process.

We chose vital signs, demographic characteristics, medical history, and chief complaint as predictive variables to develop the models. We believe these variables can represent all the information obtained during triage, and models constructed based on these variables may outperform traditional models based on subjective judgments by medical staff or patients'vital signs. To explain the models, we adopted SHapley Additive exPlanations (SHAP) [25].

This study compared the ESI five-level triage system, models based on vital signs, and ML models constructed using additional triage information, aiming to explore the maximum predictive potential of triage data for ICU admission. Additionally, it employed interpretable ML techniques to better assist clinical staff in making informed decisions during triage.

## Materials and methods

### Study design and data source

This was a retrospective cohort study, and the study design followed the TRIPOD guidelines [26]. All data were obtained from the Medical Information Mart for Intensive Care IV (MIMIC-IV, version 1.4) database. The MIMIC-IV database was used to collect the clinical information of patients who were admitted to the Beth Israel Deaconess Medical Center in Boston, Massachusetts, USA in the ED, inpatient department, and ICU from 2008 to 2019. The MIMIC-IV database is a public resource that provides global researchers with free access to clinical data. The data have been anonymized and de-identified. The database was developed by the Computational Physiology Laboratory at MIT and received approval from the Institutional Review Boards of both MIT and Beth Israel Deaconess Medical Center, exempting the requirement for informed consent from participants. After successfully completing the training course and examination of the cooperative organization (Record ID 45797033), ZL

extracted the data needed for this study on July 31, 2024. Data extraction was performed using Navicat Premium software (version 15.0), while statistical analysis and ML were conducted using R (version 4.1.3, The R Foundation for Statistical Computing, Vienna, Austria) and Python (version 3.8.5, Python Software Foundation). The main code and libraries used for the statistical analysis and ML sections in this study were detailed in the S1 Text.

## Study population and outcome

The study population included adult patients (aged ≥ 18 years) whose medical data were extracted from the MIMIC-IV ED database. The predicted outcome of this study was admission to ICU, which was extracted from the MIMIC-IV ICU database. Populations who lacked outcomes, had multiple visits, and had missing predictor variables were excluded.

## Data collection and feature engineering

We selected ESI, vital signs, demographic characteristics, medical history, and chief complaints as predictors for data collection. The vital signs included body temperature, heart rate, respiratory rate, systolic blood pressure (SBP), diastolic blood pressure (DBP), blood oxygen saturation, and pain index. The demographic characteristics included age and sex. Medical history included smoking, hypertension, diabetes, coronary heart disease, chronic heart failure, chronic obstructive pulmonary disease, chronic kidney disease, cirrhosis, and malignant tumors. For chief complaints, we selected those with a proportion greater than 1% or clinically typical severe symptoms to construct Model 3. For extracting chief complaints, we employed natural language processing techniques to address issues related to spelling errors and language inconsistencies.

In this study, we characterized continuous variables by reporting the median and interquartile range, and employed the Mann-Whitney U test to assess intergroup differences. For categorical variables, frequencies and percentages were utilized for description, with group comparisons conducted using either the chi-square test or Fisher's exact test.

For handling missing and extreme values, we excluded data points with missing predictors or outcomes. Extreme values of continuous variables were defined based on clinical plausibility and prior literature: body temperature < 34.5 °C or < 43.0 °C, SBP < 60 mmHg or > 250 mmHg, DBP < 30 mmHg or > 150 mmHg, respiratory rate < 10 times/min or > 50 times/min, blood oxygen saturation < 60% or > 100%, and heart rate < 40 beats/min or > 200 beats/min (Fig 1). Beyond these thresholds, we further assessed outliers using the interquartile range (IQR) method. Specifically, for each continuous variable, values outside 1.5 times the IQR below the first quartile (Q1) or above the third quartile (Q3) were flagged as potential outliers. These flagged values were reviewed and retained if they were clinically plausible; otherwise, they were excluded. Categorical variables were uniformly processed across all algorithms by applying one-hot encoding. This ensured consistent treatment of categorical features, regardless of the algorithm used. For example, variables such as sex (male/female) and smoking status (yes/no) were transformed into binary dummy variables, while multi-class variables such as chief complaints were expanded into multiple binary columns. To improve the accuracy of the model, we used a normalization method to scale all the variables and map the data to the [0,1] interval.

An order of priority based upon acuity utilizing the ESI Five Level triage system. This priority is assigned by a registered nurse. Level 1 is the highest priority, while level 5 is the lowest priority. The levels are:

Level 1: When Level 1 condition or patient meets ED Trigger Criteria, the triage process stops, the patient is taken directly to a room and immediate physician intervention

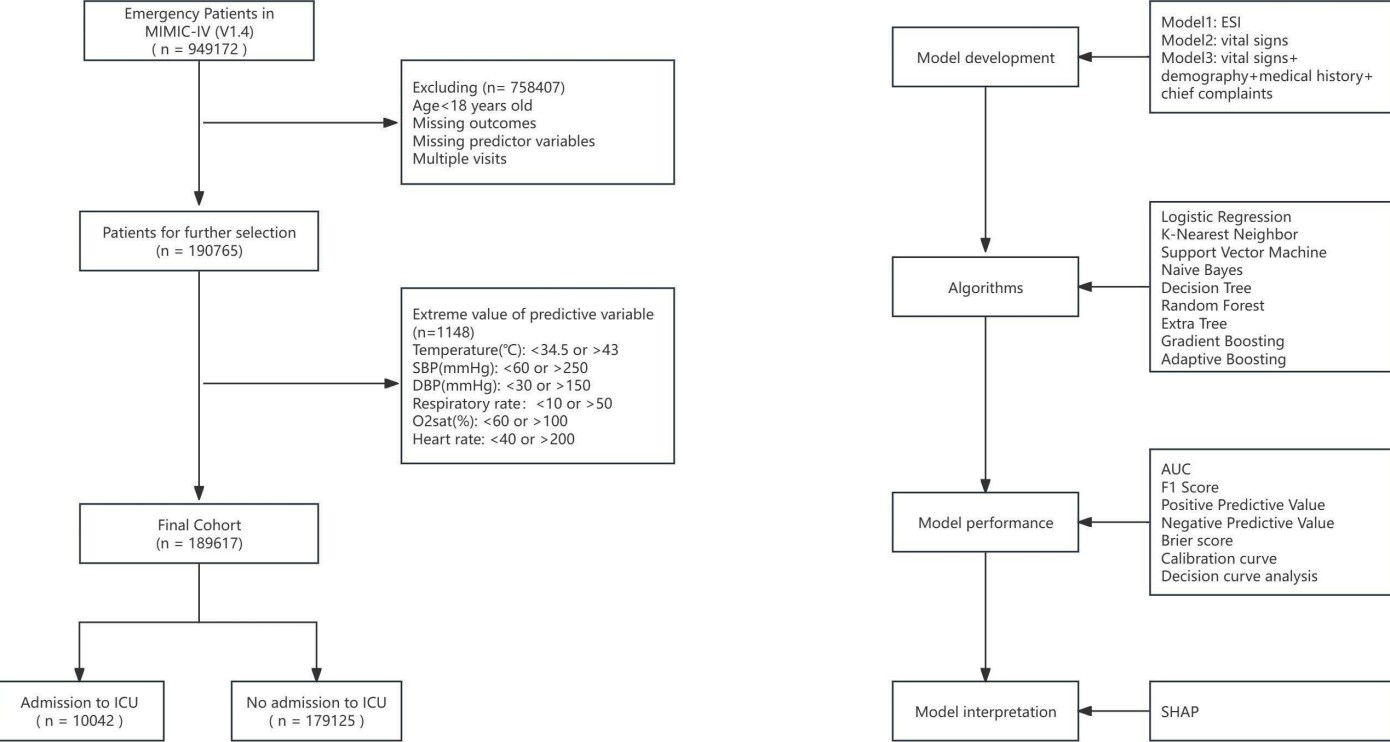

**Fig 1. Flow chart.** MIMIC, medical information mart for intensive care; SBP, systolic blood pressure; DBP, diastolic blood pressure; o2sat, oxygen saturation; ICU, intensive care unit; AUC, area under the receiver operating characteristic curve, SHAP, SHapley Additive exPlanations; ESI, Emergency Severity Index.

requested. Patient conditions which trigger level 1 include being unresponsive, intubated, apneic, pulseless, requiring a medication/intervention to alter ESI level, e.g., narcan/adenosine/cardioversion, trauma, stroke, stemi.

Level 2: When a Level 2 condition is identified, the triage nurse notifies the resource nurse and appropriate placement will be determined. Patient conditions which trigger level 2 include high risk situations, new onset confusion, suicidal/homicidal ideation, lethargy, seizures or disorientation, possible ectopic pregnancy, an immunocompromised patient with a fever, severe pain/distress, or vital sign instability.

Level 3: Includes patients requiring two or more resources (labs, EKG, x-rays, IV fluids, etc) with stable vital signs.

Level 4: Patients requiring one resource only (labs, EKG, etc)

Level 5: Patients not requiring any resources

## Resampling strategies for class imbalance

Class imbalance is a common challenge in ML, often leading to overfitting and suboptimal model performance[27]. In the current dataset, the majority class (non-ICU patients) comprises 94.69% of the data, while the minority class (ICU patients) accounts for only 5.21%. To explore the effect of addressing class imbalance on model performance, we selected the Logistic Regression (LR) algorithm as a case study and compared the results of three resampling strategies—SMOTE, SMOTE-Tomek, and RandomUnderSampler—against the original, non-resampled dataset in Models 2 and 3. Model performance was assessed using metrics

such as Precision, Sensitivity, and F1-Score, with a specific emphasis on enhancing the predictive accuracy for the minority class (ICU patients).

## Development and evaluation of the models

The data were randomly divided into a training set (80%) and test set (20%). We developed three models: Model 1 was based on the ESI five-level triage system, Model 2 was constructed using vital signs, and Model 3 was built using vital signs, demographic characteristics, medical history, and chief complaints. We applied nine algorithms to the training data to build the models: 1) LR, 2) k-Nearest Neighbor (KNN), 3) Support Vector Machine (SVM), 4) naive Bayes, 5) Decision Tree, 6) Random Forest, 7) Extra Tree, 8) Gradient Boosting, and 9) Adaptive Boosting (AdaBoost). By tuning the regularization parameters, we aimed to control model complexity and enhance its generalization ability and performance. Model 1, containing only the ESI variable, used the LR algorithm. Models 2 and 3 employed all nine algorithms to explore the best area under the receiver operating characteristic curve (AUC) value. The AUC and its 95% confidence interval were determined using the DeLong method. We evaluated the performance of the models on the test data using several metrics, including the F1 Score, Area Under the Precision-Recall Curve (AUC-PR), Positive Predictive Value (PPV), Negative Predictive Value (NPV), Brier score, calibration curve, and decision curve analysis. We utilized the SHAP method to interpret different algorithms and assessed the risk of ICU admission to better assist healthcare providers in decision-making.

## Results

### Characteristics of study sample

A total of 189,167 emergency patients were included in this study, of which 10,042 (5.31%) were admitted to the ICU and 17,9125 (94.69%) were not admitted to ICU (Table 1). The correlation coefficient between SBP and DBP was 0.5, while it was less than 0.25 for the other continuous variables. Body temperature, heart rate, respiratory rate, and age exhibited positive correlations with the outcome, whereas SBP, DBP, blood oxygen saturation, and pain index showed negative correlations with the outcome (Fig 2A). Restricted cubic splines illustrated the nonlinear relationships between systolic and diastolic blood pressure and the outcome, whereas temperature, heart rate, respiratory rate, blood oxygen saturation, pain index, and age displayed linear relationships with the outcome (Fig 2B). Bar charts illustrating categorical variables with outcomes were presented in Fig 2C.

### Effect of resampling on the dataset

The results showed that, compared to the original dataset without resampling, all three resampling techniques—SMOTE, SMOTE-Tomek, and RandomUnderSampler—significantly improved the Precision, Sensitivity, and F1-Score for the minority class, while their AUC values were largely consistent (S1 Table). Among them, RandomUnderSampler, which uses only actual data, minimizes the risk of overfitting that can arise from generating synthetic samples, while also being less computationally demanding. Considering the large scale of the dataset and the need for multi-model comparisons in this study, random undersampling was chosen as the primary strategy to mitigate class imbalance, as it offers an optimal trade-off between predictive performance and computational efficiency.

### Performance of the models

As shown in Fig 3, when Model 1 was used for prediction, the AUC value was 0.68. When Model 2 was used for prediction, the AUC values of the nine algorithms, in descending order,

**Table 1. Baseline analysis of variables and outcomes.**

| Variable | Non-ICU | ICU | P-value |
|---|---|---|---|
| | (n = 179125) | (n = 10042) | |
| **Emergency Severity Index** | | | |
| 1 | 8,732 (4.9%) | 2,051 (20%) | |
| 2 | 88,497 (49%) | 6,198 (62%) | |
| 3 | 80,489 (45%) | 1,789 (18%) | |
| 4 | 1,365 (0.8%) | 4 (<0.1%) | |
| 5 | 42 (<0.1%) | 0 (0%) | |
| **Vital signs in triage** | | | |
| Temperature (°C) | 36.7 (36.4, 37.0) | 36.7 (36.4, 37.2) | <0.001 |
| Heart rate (beats/min) | 84 (73, 98) | 92 (78, 108) | <0.001 |
| Respiratory rate (times/min) | 18 (16, 18) | 18 (16, 20) | <0.001 |
| Oxygen saturation (%) | 98 (97, 100) | 97 (95, 99) | <0.001 |
| Systolic blood pressure (mmHg) | 133 (119, 150) | 122 (105, 141) | <0.001 |
| Diastolic blood pressure (mmHg) | 76 (66, 86) | 69 (58, 80) | <0.001 |
| Pain index | 3 (0, 7) | 0 (0, 7) | <0.001 |
| **Demographic characteristics** | | | |
| Age (year) | 58 (43, 71) | 66 (55, 77) | <0.001 |
| Male [n(%)] | 87087 (48.6%) | 5533 (55.1%) | <0.001 |
| **Medical history [n(%)]** | | | |
| Smoke | 34539 (19.3%) | 2941 (29.3%) | <0.001 |
| Chronic obstructive pulmonary disease | 975 (0.5%) | 174 (1.7%) | <0.001 |
| Chronic kidney diseases | 25032 (14.0%) | 2673 (26.6%) | <0.001 |
| Coronary heart disease | 30948 (17.3%) | 2820 (28.1%) | <0.001 |
| Chronic heart failure | 5357 (3.0%) | 792 (7.9%) | <0.001 |
| Diabetes | 45571 (25.4%) | 3673 (36.6%) | <0.001 |
| Tumor | 5174 (2.9%) | 454 (4.5%) | <0.001 |
| Hypertension | 31494 (17.6%) | 1842 (18.3%) | 0.052 |
| Cirrhosis | 5731 (3.2%) | 1026 (10.2%) | <0.001 |
| **Chief complaints [n(%)]** | | | |
| Fever[§] | 8893 (5.0%) | 1006 (10.0%) | <0.001 |
| Chills | 371 (0.2%) | 31 (0.3%) | 0.031 |
| Tachycardia | 1076 (0.6%) | 141 (1.4%) | <0.001 |
| Weakness[§] | 8087 (4.5%) | 660 (6.6%) | <0.001 |
| Anemia | 1262 (0.7%) | 86 (0.9%) | 0.078 |
| Diarrhea[§] | 2045 (1.1%) | 109 (1.1%) | 0.605 |
| Dizziness[§] | 3909 (2.2%) | 109 (1.1%) | <0.001 |
| Dyspnea[§] | 12915 (7.2%) | 1312 (13.1%) | <0.001 |
| Headache[§] | 3486 (1.9%) | 124 (1.2%) | <0.001 |
| Fatigue | 766 (0.4%) | 31 (0.3%) | 0.073 |
| Low blood pressure[§] | 1127 (0.6%) | 448 (4.5%) | <0.001 |
| High blood pressure | 785 (0.4%) | 18 (0.2%) | <0.001 |
| Lethargy[§] | 959 (0.5%) | 211 (2.1%) | <0.001 |
| Jaundice[§] | 782 (0.4%) | 126 (1.3%) | <0.001 |
| Pneumonia | 457 (0.3%) | 125 (1.2%) | <0.001 |
| Syncope[§] | 3381 (1.9%) | 105 (1.0%) | <0.001 |
| Abscess | 762 (0.4%) | 42 (0.4%) | 0.915 |
| Mantal altered[§] | 3714 (2.1%) | 680 (6.8%) | <0.001 |

*(Continued)*

**Table 1.** (Continued)

| Variable | Non-ICU | ICU | P-value |
|---|---|---|---|
| | (n = 179125) | (n = 10042) | |
| Swelling§ | 4362 (2.4%) | 165 (1.6%) | <0.001 |
| Anxiety | 1100 (0.6%) | 6 (0.1%) | <0.001 |
| Palpitations§ | 1505 (0.8%) | 23 (0.2%) | <0.001 |
| Hemoptysis | 429 (0.2%) | 79 (0.8%) | <0.001 |
| Aphasia | 137 (0.1%) | 4 (0.0%) | 0.190 |
| Numbness | 1553 (0.9%) | 15 (0.1%) | <0.001 |
| Edema | 388 (0.2%) | 19 (0.2%) | 0.564 |
| Hematuria | 1024 (0.6%) | 54 (0.5%) | 0.660 |
| Nausea or Vomiting§ | 9808 (5.5%) | 490 (4.9%) | 0.010 |
| Bradycardia | 342 (0.2%) | 22 (0.2%) | 0.531 |
| Cough§ | 3564 (2.0%) | 251 (2.5%) | <0.001 |
| Confusion§ | 1861 (1.0%) | 153 (1.5%) | <0.001 |
| Slurred | 644 (0.4%) | 34 (0.3%) | 0.732 |
| Sore throat | 766 (0.4%) | 30 (0.3%) | 0.052 |
| Dysuria | 816 (0.5%) | 24 (0.2%) | 0.001 |
| Cellulitis | 802 (0.4%) | 19 (0.2%) | <0.001 |
| Hematemesis | 442 (0.2%) | 133 (1.3%) | <0.001 |
| Hyperglycemia§ | 1559 (0.9%) | 125 (1.2%) | <0.001 |
| Hypoglycemia | 505 (0.3%) | 35 (0.3%) | 0.223 |
| Seizure§ | 1844 (1.0%) | 115 (1.1%) | 0.265 |
| Rash | 716 (0.4%) | 26 (0.3%) | 0.028 |
| Gastrointestinal bleeding§ | 1264 (0.7%) | 168 (1.7%) | <0.001 |
| Stroke | 81 (0.0%) | 6 (0.1%) | 0.509 |
| Slurred speech | 638 (0.4%) | 34 (0.3%) | 0.773 |
| Gait unsteady | 643 (0.4%) | 23 (0.2%) | 0.032 |
| Visual changes | 626 (0.3%) | 6 (0.1%) | <0.001 |
| Wound§ | 3409 (1.9%) | 141 (1.4%) | <0.001 |
| Lightheaded | 779 (0.4%) | 26 (0.3%) | 0.008 |
| Fall§ | 8533 (4.8%) | 370 (3.7%) | <0.001 |
| Urinary retention | 548 (0.3%) | 32 (0.3%) | 0.822 |
| Chest pain§ | 15351 (8.6%) | 424 (4.2%) | <0.001 |
| Abdominal pain§ | 24388 (13.6%) | 1118 (11.1%) | <0.001 |
| Back pain§ | 4846 (2.7%) | 166 (1.7%) | <0.001 |
| Leg pain§ | 2422 (1.4%) | 69 (0.7%) | <0.001 |
| Flank pain§ | 1893 (1.1%) | 63 (0.6%) | <0.001 |
| Arm pain | 964 (0.5%) | 24 (0.2%) | <0.001 |
| Arthralgia pain§ | 4516 (2.5%) | 90 (0.9%) | <0.001 |
| Limb pain§ | 4890 (2.7%) | 136 (1.4%) | <0.001 |

§Chief complaints included in Model 3.

ICU, intensive care unit.

were as follows: Gradient Boosting (0.76), naive Bayes (0.75), LR (0.74), Random Forest (0.71), AdaBoost (0.69), Extra Tree (0.67), KNN (0.61), Decision Tree (0.55), and SVM (0.33). When Model 3 was used for prediction, the AUC values of the nine algorithms, in descending order, were as follows: Gradient Boosting (0.81), LR (0.81), naive Bayes (0.80), Random Forest

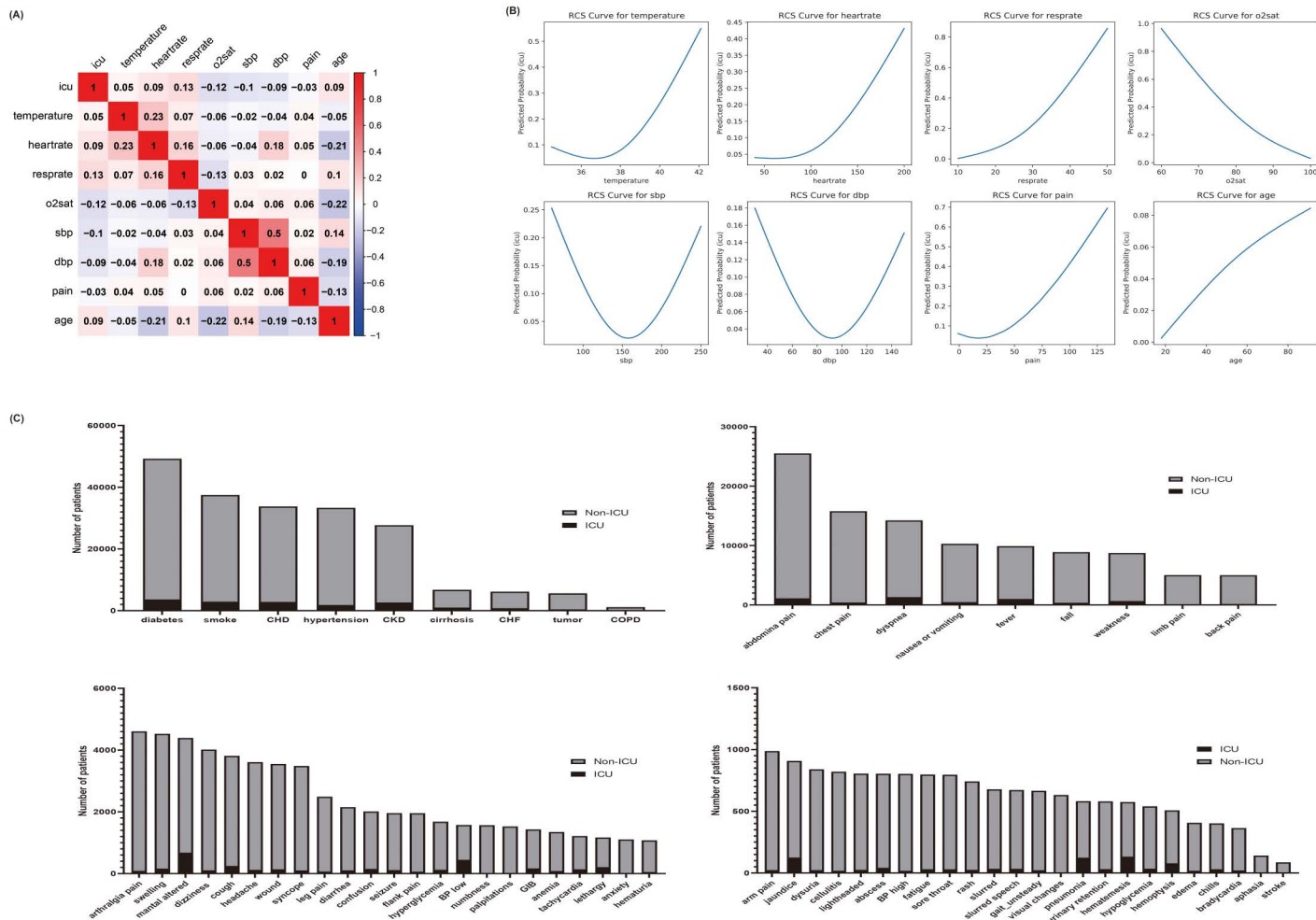

**Fig 2. Visualization of the relationship between variables and outcomes.** A: Heatmap of correlation between continuous variables. B: Restricted cubic spline analysis of the linear relationship between continuous variables and outcomes. C: Bar chart of the relationship between categorical variables and outcomes. ICU, intensive care unit; o2sat, oxygen saturation; SBP, systolic blood pressure; DBP, diastolic blood pressure; COPD, chronic obstructive pulmonary disease; CKD, chronic kidney diseases; CHF, chronic heart failure; CHD, coronary heart disease; GIB, gastrointestinal bleeding.

(0.80), AdaBoost (0.75), Extra Tree (0.73), Decision Tree (0.65), KNN (0.63), and SVM (0.35). LR, Gradient Boosting, naive Bayes, and Random Forest algorithms were compared for Model 3, while other models and algorithms were not further evaluated due to their lower AUC values. Table 2 presented metrics evaluating the overall performance, discrimination, calibration, and clinical utility of these four models. Calibration curves for the algorithms were shown in Fig 4A, decision curve analysis in Fig 4B. Overall, after comparison, Model 3 demonstrated superior predictive performance compared to Models 1 and 2, with LR, Random Forest, and Gradient Boosting being the three best-performing algorithms within Model 3.

## Feature importance and interpretation of the models

The feature importance for the three best-performing ML algorithms in Model 3 (LR, Random Forest, Gradient Boosting) was detailed in the S1 Fig. Using Case16 as an example, we analyzed the predictions of these three algorithms using the SHAP method and provided the probability of ICU admission (Fig 5). The predicted probabilities of sepsis

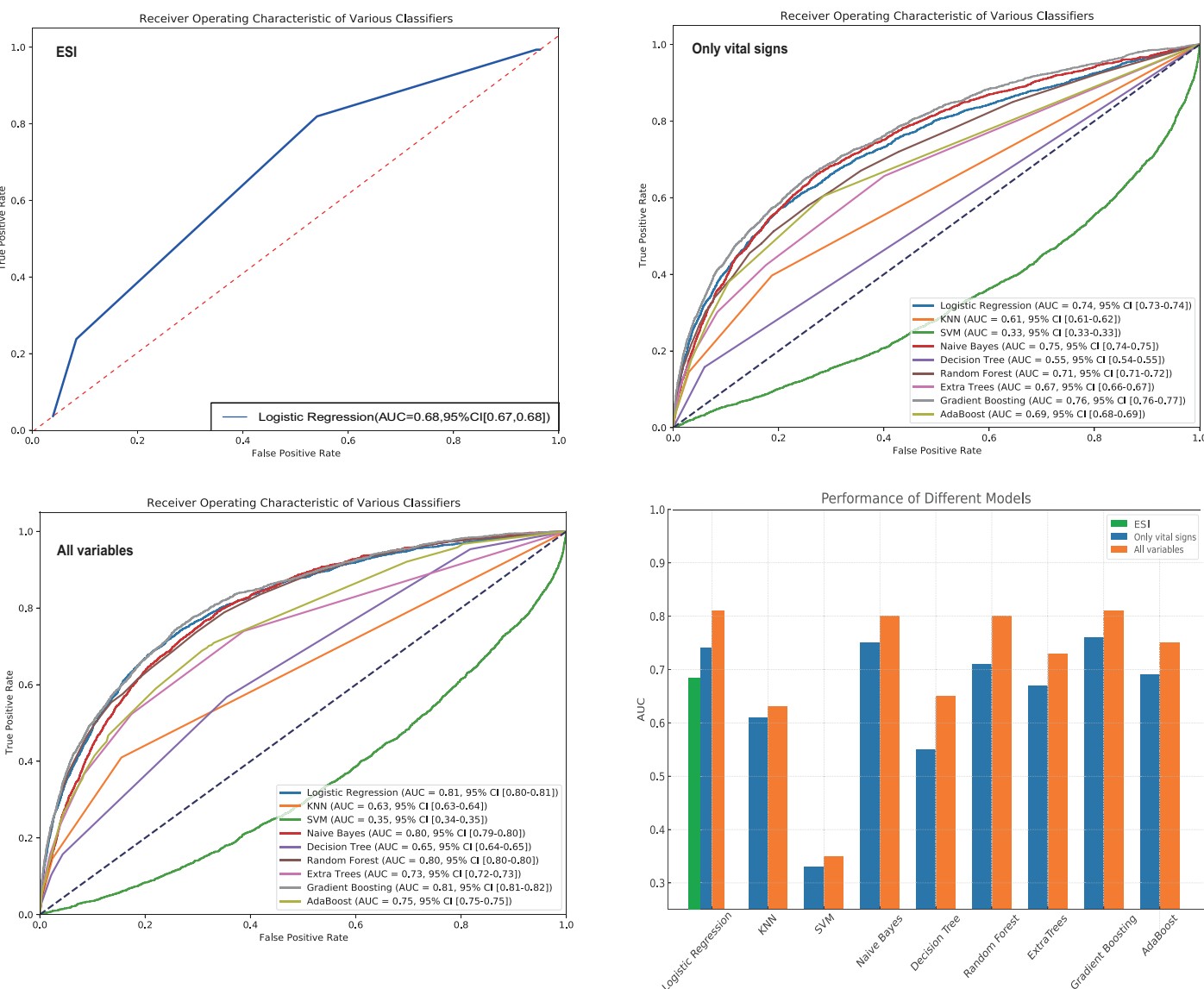

**Fig 3. AUC values of the nine algorithms under three models.** ESI, Emergency Severity Index; AUC, area under the receiver operating characteristic curve; KNN, k-Nearest Neighbor; SVM, Support Vctor Machine; AdaBoost, Adaptive Boosting.

using the three algorithms were as follows: LR (77%), Random Forest (84%), and Gradient Boosting (96%).

In the SHAP method, f (X) represented the final prediction result, which equaled the baseline value E [f (X)] plus the sum of all variable SHAP values. The SHAP values quantified the quantity and direction of each variable's influence on predicting the outcome. Blue and red respectively represented decreases or increases in risk, with longer arrows indicating greater effects. The baseline value E [f (X)] was equivalent to the average risk in the dataset.

## Discussion

In this study, we predicted the risk of ICU admission based on triage information from 189,716 emergency patients by constructing interpretable ML models. Our study indicated

**Table 2. Performance of four models.**

| Models | Overall | | | | Discrimination | Calibration | Clinical Usefulness |
|---|---|---|---|---|---|---|---|
| | F1 Score | AUC-PR | PPV | NPV | AUC(95%CI) | Brier score | Net benefit at threshold of 5% |
| LR | 0.72 | 0.79 | 0.73 | 0.72 | 0.81(0.80-0.81) | 0.045 | 0.47 |
| Naive Bayes | 0.68 | 0.92 | 0.69 | 0.69 | 0.80(0.79-0.80) | 0.199 | 0.38 |
| Random Forest | 0.73 | 0.79 | 0.72 | 0.73 | 0.80(0.80-0.80) | 0.045 | 0.47 |
| Gradient Boosting | 0.74 | 0.80 | 0.74 | 0.72 | 0.81(0.81-0.82) | 0.044 | 0.47 |

AUC, area under the receiver operating characteristic curve; AUC-PR, area under the precision-recall curve; LR, Logistic Regression; PPV, positive predictive value; NPV, negative predictive value.

that adding demographic characteristics, medical history, and chief complaints to the triage prediction model improved the predictive performance for critical conditions compared to using only vital signs or the ESI five-level triage system for prediction. Furthermore, interpretable ML helped by providing transparency and insight into predictions, aiding medical decision-making during triage.

We visualized the data to intuitively observe the relationship between various characteristic variables and outcomes. In this study, all vital signs were continuous variables. Restricted cubic splines demonstrated significant linear or U-shaped relationships between these variables and outcomes (Fig 2B). Traditionally, triage warning scores categorize vital signs, but we retained them as continuous variables to explore maximum predictive efficacy, which categorical variables often diminish. Unfortunately, the MIMIC-IV ED database lacked data on patient conscious state, which was only reflected in the chief complaint. We displayed relationships between classification variables and outcomes as columns, listed in Fig 2C. Descriptive statistics indicated that most variables correlated with the occurrence of the outcome.

In our study, the AUC value of the ESI five-level triage system was 0.68, which was lower than that of Models 2 and 3. Previous studies have reported that the ESI has lower sensitivity in accurately identifying critically ill patients, which may lead to delays in care [28,29]. As shown in Fig 3, after adding additional variables, the AUC values of most of the algorithms were improved. This showed that, based on sEMR, collecting additional variables during triage can improve the prediction efficiency, at least in terms of ICU admission. The popularization of sEMR has become a trend in the medical system, providing a carrier for artificial intelligence to serve medical care [30,31]. Among the nine algorithms, the highest AUC value was 0.81, which was based on LR and Gradient Boosting, followed by 0.80, which was based on naive Bayes and Random Forest. Although sEMR is quite convenient, its ability to collect my information in the triage setting is limited and typically does not extend beyond four aspects: vital signs, medical history, demographics, and chief complaint. Therefore, we believe that an AUC of 0.81 can represent the maximum predictive performance for triage in predicting ICU admission.

According to the AUC values, we further compared the prediction effects of LR, Gradient Boosting, naive Bayes, and Random Forests because the AUC value is the most important indicator for evaluating the prediction model [32]. The AUC value was used to judge the discrimination ability of the model, whereas the calibration curve was used to judge the degree of conformity between the prediction and the actual situation. A perfect calibration curve should be on the 45-degree line [33], and in this study, the naive Bayes algorithm performed poorly, while the remaining three algorithms performed excellently, as shown in Fig 4A. The decision curve analysis showed that the performance of naive Bayes was noticeably inferior to the other three algorithms (Fig 4B). As shown in Table 2, considering the evaluation metrics

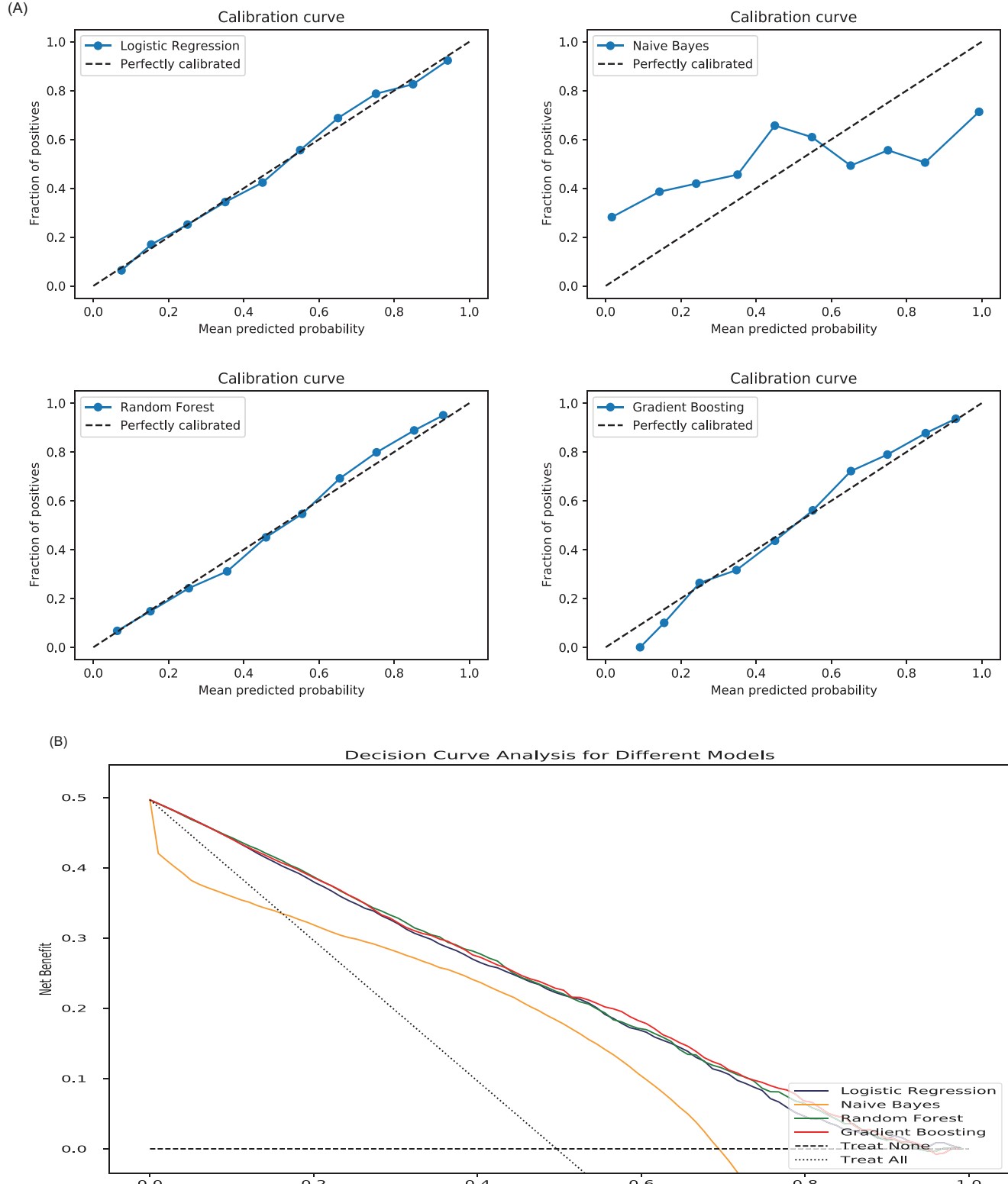

**Fig 4. The performance of the four algorithms.** A: Calibration curves of the four algorithms. B: Decision curve analysis curves of the four algorithms.

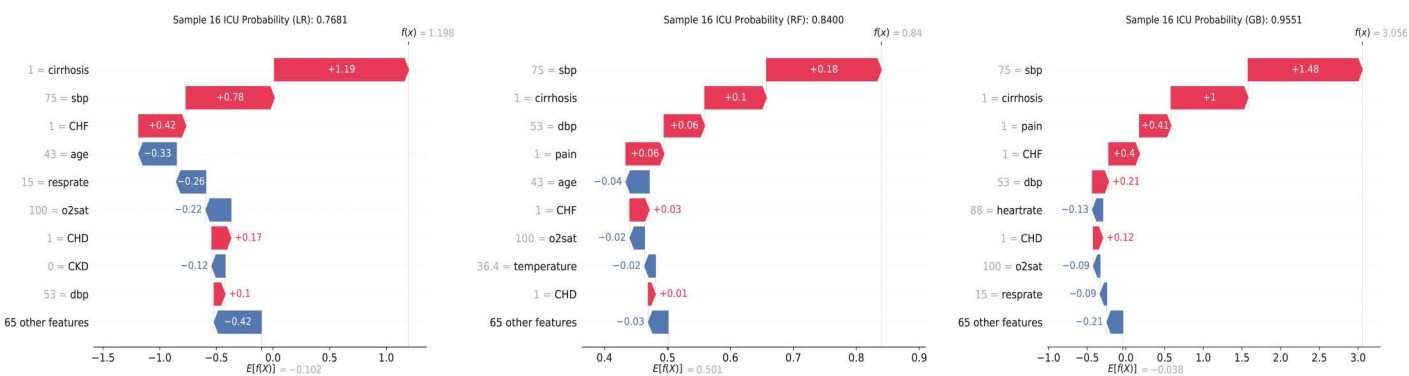

**Fig 5. Interpretation of the models.** LLR, Logistic Regression; RF, Random forest; GB, Gradient Boosting; sbp, systolic blood pressure; dbp, diastolic blood pressure; o2sat, oxygen saturation; CKD, chronic kidney diseases; CHD, coronary heart disease; CHF, chronic heart failure.

of overall performance, discriminative ability, calibration ability, and clinical application value, Gradient Boosting performed slightly better than LR and Random Forest, while naive Bayes performed the worst. Although the improvement in AUC by complex models such as Gradient Boosting over simpler ones like Logistic Regression appears marginal (0.81 vs. 0.80), the added complexity is justifiable in certain contexts. The AUC is only one evaluation metric and does not fully capture a model's overall performance. Gradient Boosting achieved the highest F1 Score (0.74) among all models, along with the lowest Brier score (0.044), indicating superior overall performance, calibration, and probabilistic prediction accuracy, as well as comparable or better clinical usefulness (Table 2, Fig 4B). Additionally, complex models can capture non-linear relationships and interactions between variables that simpler models may fail to identify, which is particularly relevant in clinical datasets with inherent heterogeneity. However, we acknowledge that simpler models like Logistic Regression remain highly interpretable, computationally efficient, and better suited for resource-limited or transparent decision-making scenarios. Therefore, the choice between simple and complex models requires balancing the slight gain in performance against the trade-offs in interpretability and computational demands depending on the clinical application.

Many studies have concluded that ML can deal better with high-order nonlinear interactions between variables than LR when faced with complex data or numerous features [34–36]. Indeed, some studies have found that ML is better than LR at predicting critical illness or poor prognosis [37–40]; however, a systematic review in 2019 showed that ML had no performance advantage over LR for clinical prediction models [23]. Yun et al. [41] compared the ML model and KTAS to predict the risk of critical care outcomes using patient triage data. The results showed that the Gradient Boosting algorithm was superior to KTAS, and the AUC of Gradient Boosting was 0.86. Yun et al. included more variables than we did, such as mode of arrival, interval between onset and arrival, and state of consciousness, but did not include medical history. De Hond et al. [42] added laboratory tests within 2h to predict the risk of hospitalization in emergency patients. The results showed that ML had excellent predictive performance, similar to LR, for predicting hospitalization rate, and the best performing ML algorithm was Gradient Boosting. The advantage of the Gradient Boosting algorithm is that it can identify a variety of distinguishing features and feature combinations.

This study exemplified Case16 to explain the three better-performing algorithms: LR, Random Forest, and Gradient Boosting. Although different algorithms included slightly different variables and weights, they all could assist healthcare professionals in clinical judgment. In our study, all three algorithms included vital signs, demographics, medical history, and chief

complaints in the model (Fig 5). This also demonstrated that using only vital signs for modeling in the triage alert model may not be optimal. SHAP fairly allocates contribution values for each feature in the sample, ultimately explaining the difference between the predicted value of an individual sample and the average predicted value of the model [43]. SHAP also has some drawbacks, such as long computation times when explaining certain ML algorithms, which limits its clinical applicability. For instance, in this study, the computation time for Random Forest was significantly longer than for LR and Gradient Boosting.

The primary significance of this study lies in our demonstration that incorporating more triage information into the model development process was superior to traditional early warning models. Additionally, we employed nine different algorithms for prediction and identified the optimal one through comparison. As sEMR continues to develop, these results may provide more options for future triage warning methods, surpassing the sole reliance on vital signs or subjective judgment. At the same time, we utilized interpretable ML methods, enabling ML to assist healthcare professionals in clinical decision-making, thereby providing feasibility and credibility for the application of this new warning model in triage. For the first time, we provided the probability of patients being admitted to the ICU using interpretable ML methods. By assessing these probabilities, triage healthcare personnel could further determine the severity of a patient's condition. However, the ethical implications of using automated decision-making systems in clinical practice warrant careful consideration. While such systems can enhance the accuracy and efficiency of triage processes, they must not replace human judgment but rather serve as a supportive tool for healthcare professionals. Over-reliance on automated systems may risk overlooking nuanced clinical contexts that are difficult to quantify or encode into models. Additionally, the potential for algorithmic bias, stemming from imbalanced datasets or unrepresentative training data, must be addressed to ensure equitable care across diverse patient populations. Transparency in model development, validation, and implementation is essential to build trust among healthcare providers and patients. Lastly, the deployment of such systems should always prioritize patient autonomy, ensuring that automated predictions are used to inform, rather than dictate, clinical decisions.

The limitations of this study mainly manifest in the following aspects. Firstly, as a single-center study, external validation was not possible. Secondly, as a retrospective study, it was unable to include all potential variables, such as consciousness status and history of alcohol abuse. Certain chief complaints were not very precise; for example, some patients'chief complaints were "low blood pressure" or "low blood sugar". If their proportion exceeded 1%, we included these less standardized chief complaints. Thirdly, removing patients with missing and extreme values may introduce selection bias, particularly if the data were not missing completely at random. For example, extreme values could carry clinically important information, and their exclusion might affect model performance. Future studies should consider applying advanced imputation methods, such as multiple imputation by chained equations (MICE), to address missing values and compare the results with models based on complete-case data to evaluate potential biases. Additionally, the clinically defined thresholds for extreme values were conservative, which might have excluded meaningful data points, and future efforts could explore machine learning-based anomaly detection methods or use winsorization to better account for the impact of extreme values on model performance. Fourthly, in terms of population selection, we excluded patients from the MIMIC-IV ED who lacked outcomes and those with multiple visits. Since these patients constituted a significant proportion, there may be some selection bias. Finally, while our predictive model provided the probability of ICU admission, it did not directly determine the severity of emergency patient conditions, and further research is needed to confirm whether it can improve clinical decision-making and better stratify emergency patients by severity.

## Conclusions

In conclusion, this study developed an interpretable ML triage warning model based on vital signs, demographics, medical history, and chief complaints. It was more effective in predicting whether patients needed ICU admission compared to traditional early warning models based on vital signs or ESI. Interpretable ML can assist healthcare professionals in making clinical decisions during triage.

## Supporting information

**S1 Text. Primary code used in this study.**
(PDF)

**S1 Table. Performance comparison of ML model (LR) before and after resampling.**
(PDF)

**S1 Fig. Feature importance in Logistic Regression, Random Forest and Gradient Boosting model.**
(PDF)

## Acknowledgements

The authors would like to express their sincere gratitude to the MIMIC database for providing publicly available data, which has been instrumental in facilitating this research.

## Author contributions

**Conceptualization:** Zheng Liu, Wenqi Shu, Wei Chong.

**Data curation:** Zheng Liu.

**Formal analysis:** Zheng Liu.

**Funding acquisition:** Zheng Liu.

**Investigation:** Wenqi Shu.

**Methodology:** Zheng Liu, Wei Chong.

**Project administration:** Wenqi Shu, Hongyan Liu, Xuan Zhang.

**Resources:** Wei Chong.

**Software:** Zheng Liu.

**Supervision:** Wei Chong.

**Validation:** Hongyan Liu, Xuan Zhang.

**Visualization:** Zheng Liu.

**Writing – original draft:** Zheng Liu, Wenqi Shu.

**Writing – review & editing:** Zheng Liu, Wenqi Shu.

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
