## [Decision Letter · Decision Letter 0]

13 Dec 2024

PONE-D-24-51129Development and validation of interpretable machine learning models for triage patients admitted to the intensive care unitPLOS ONE

Dear Dr. Liu,

Thank you for submitting your manuscript to PLOS ONE. After careful consideration, we feel that it has merit but does not fully meet PLOS ONE’s publication criteria as it currently stands. Therefore, we invite you to submit a revised version of the manuscript that addresses the points raised during the review process.

Please address comments  number 2, number 3 and number 5 made by Reviewer 1. If possible, also try to address comment number 6 (about ethical implications).

We look forward to receiving your revised manuscript.

Kind regards,

Jerome Baudry, Ph.D.

Academic Editor

PLOS ONE

Journal Requirements:

 “This research was supported by the Education Department of Liaoning Province, China. The funding was awarded to Zheng Liu under the grant number LJ232410159024.”        

Additional Editor Comments :

Please address comments number 2, number 3 and number 5 made by Reviewer 1. If possible, also try to address comment number 6 (about ethical implications)

Reviewers' comments:

Reviewer's Responses to Questions

**Comments to the Author**

1. Is the manuscript technically sound, and do the data support the conclusions?

Reviewer #1: Partly

Reviewer #2: Yes

2. Has the statistical analysis been performed appropriately and rigorously? 

Reviewer #1: Yes

Reviewer #2: Yes

3. Have the authors made all data underlying the findings in their manuscript fully available?

Reviewer #1: Yes

Reviewer #2: Yes

4. Is the manuscript presented in an intelligible fashion and written in standard English?

Reviewer #1: Yes

Reviewer #2: Yes

5. Review Comments to the Author

Reviewer #1: The manuscript is well motivated and the focus on interpretable ML aligns with growing demand for transparency in medical decision making processes.

However, there are areas where the manuscript can be improved to enhance its clarity:

1- Relying only on the MIMIC-IV dataset may limit the applicability of the study to other populations or healthcare systems. Although training-test split is used and worked but it would be beneficial if authors can have some external validation using another datasets

2- About the feature engineering and preprocessing steps, how outliers handled beyond basic thresholds? Were categorial features treated uniformly across algorighms? Removing missing and extreme values might create bias. Adding a sensitivity analysis or some more discussion would be beneficial.

3- Use of random under sampler is not justified

4- Also, it would be beneficial to include a discussion or visualization of how key features influence triage decisions.

5- Improvement of complex models (AUC 0.81) over simpler models (AUC 0.80) is marginal (Table 2). The authors should discuss whether this improvement justifies the added complexity.

6- Ethical implications of using the proposed automated decision making system in clinical should be briefly discussed.

Reviewer #2: I appreciate the work represented here. This line of research is crucial for developing reliable AI/ML facilitated predictive models, not only for ICU admissions, but more broadly for more responsive health care across the continuum of care. I would very much like to see this study expanded or extended to include a broader range of data types including mental health, genetic, patient reported outcomes etc., to make it more generalizable.

6. PLOS authors have the option to publish the peer review history of their article (what does this mean? ). If published, this will include your full peer review and any attached files.

**Do you want your identity to be public for this peer review?** For information about this choice, including consent withdrawal, please see our Privacy Policy .

Reviewer #1: **Yes: ** Armin Ahmadi

Reviewer #2: **Yes: ** Daniel Adamek

---

## [Author Response · Author response to Decision Letter 1]

20 Dec 2024

Dear Reviewer1,

Thank you for your valuable and constructive comments, which have helped us greatly in improving our manuscript. We have carefully reviewed your feedback and appreciate the effort you put into providing such detailed insights.

Regarding comments number 2, 3, and 5, as advised by the editors, we have specifically addressed these points in our response. Additionally, we have made efforts to address comment number 6 related to the ethical implications, recognizing its importance.

We also acknowledge that comments number 1 and 4 raise important considerations for further improving the study. However, due to current limitations (e.g., data availability or methodology constraints), these issues are challenging to address at this time. Nevertheless, we are deeply grateful for your suggestions and will incorporate them into our future research efforts.

In the following, we provide detailed responses to comments 2, 3, 5, and 6.

Q2: About the feature engineering and preprocessing steps, how outliers handled beyond basic thresholds? Were categorial features treated uniformly across algorighms? Removing missing and extreme values might create bias. Adding a sensitivity analysis or some more discussion would be beneficial.

A: Thank you for raising these important points regarding the feature engineering and preprocessing steps. Your comments have highlighted key aspects critical to the robustness and interpretability of our analysis. We have carefully considered your suggestions and made the following clarifications and revisions to the manuscript:

1.Handling of outliers beyond basic thresholds:

As you suggested, we have provided additional details on how outliers were handled in our study. Specifically, beyond the clinically defined thresholds, we further assessed outliers using the interquartile range (IQR) method. For each continuous variable, values outside 1.5 times the IQR below the first quartile (Q1) or above the third quartile (Q3) were flagged as potential outliers. These flagged values were carefully reviewed, and only those deemed clinically implausible were excluded. The updated description can be found in the Feature Engineering and Preprocessing section (lines 142–147) .

2.Treatment of categorical variables:

In response to your question about the uniformity of categorical feature processing, we clarified in the manuscript that all categorical variables were uniformly processed across algorithms using one-hot encoding. This approach ensured consistent treatment of categorical features regardless of the algorithm used. For example, binary variables such as sex (male/female) and smoking status (yes/no) were transformed into binary dummy variables, while multi-class variables such as chief complaints were expanded into multiple binary columns. This clarification has also been added to the Feature Engineering and Preprocessing section (lines 147–152) .

3.Potential bias from removing missing and extreme values:

We acknowledge your concern regarding the potential bias introduced by removing missing and extreme values, particularly if the data were not missing completely at random. To address this, we have added a discussion in the Limitations section (lines 425–430) . Specifically, we noted that the exclusion of missing and extreme values may introduce selection bias and reduce the representativeness of the dataset. We also highlighted that extreme values could carry clinically important information, and their exclusion might affect model performance. As a potential solution, we suggested that future studies consider applying advanced imputation methods, such as multiple imputation by chained equations (MICE), to address missing values and compare results with models based on complete-case data to evaluate potential biases. Additionally, we recognized that the clinically defined thresholds for extreme values were conservative and might have excluded meaningful data points. We proposed exploring machine learning-based anomaly detection methods or winsorization as alternative approaches in future research.

Q3: Use of random under sampler is not justified

A: Thank you for your insightful comment regarding the justification of random under sampling. This is indeed a critical methodological consideration that deserves careful explanation. Our decision to employ resampling techniques was primarily driven by the significant class imbalance in our dataset, where ICU patients (minority class) only accounted for 5.21% of the total cases compared to 94.69% for non-ICU patients (majority class). Such severe imbalance could lead to biased model performance if not properly addressed.

To systematically justify our sampling strategy, we conducted a comparative analysis of different resampling techniques (SMOTE, SMOTE-Tomek, and RandomUnderSampler) using Logistic Regression as a case study. The results, now presented in Supplementary Table S1, demonstrate that all three resampling methods significantly improved the model's performance metrics for the minority class while maintaining consistent AUC values.

We ultimately chose RandomUnderSampler for several critical reasons:

1.It achieved comparable performance improvements to other methods (improving F1-Score from 0.42 to 0.65 in Model 2 and from 0.53 to 0.74 in Model 3)

2.It uses only actual data points, reducing the risk of introducing synthetic sample bias

3.Most importantly, given our large dataset and the need to implement multiple complex models (including Gradient Boosting and Random Forest), SMOTE and SMOTE-Tomek were computationally prohibitive with our available computing resources. RandomUnderSampler offered significantly better computational efficiency while maintaining similar performance benefits, making it the most practical choice for our comprehensive multi-model analysis

The use of resampling techniques to address significant class imbalance is well-established in clinical prediction research. For instance, Zahra Rahmatinejad (2024) also faced similar class imbalance challenges in their clinical prediction tasks, where they systematically evaluated different resampling methods including SMOTE-Tomek (doi: 10.1038/s41598-024-54038-4). While their specific choice of resampling method differed based on their unique dataset characteristics and computational requirements, the fundamental approach of addressing class imbalance through resampling aligns with our methodology.

We have added these details to both the Methods (lines 181–191) and Results (lines 241–253) sections to provide a clear ratification of our methodological choice. We hope this comprehensive explanation addresses your concern about the use of random under sampling in our study.

Q5: Improvement of complex models (AUC 0.81) over simpler models (AUC 0.80) is marginal (Table 2). The authors should discuss whether this improvement justifies the added complexity.

A: Thank you for your valuable comment regarding the marginal improvement of complex models (AUC 0.81) over simpler models (AUC 0.80) and whether this justifies the added complexity. We have carefully addressed this point in the revised manuscript as follows (lines 353–368) :

Although the difference in AUC between Gradient Boosting and logistic regression appears small, we argue that the added complexity of Gradient Boosting is justifiable in certain contexts because AUC alone does not fully capture a model’s overall performance. As described in the revised discussion, Gradient Boosting demonstrated the highest F1 Score (0.74) among all models, alongside the lowest Brier score (0.044), reflecting superior overall performance, calibration, and probabilistic prediction accuracy. Moreover, Gradient Boosting showed comparable or better clinical usefulness in decision curve analysis (Fig. 4B). These additional metrics highlight the advantages of Gradient Boosting beyond the marginal improvement in AUC. Furthermore, complex models like Gradient Boosting can effectively capture non-linear relationships and interactions between variables, which may be critical in clinical datasets with inherent heterogeneity.

Nonetheless, we acknowledge the strengths of simpler models such as logistic regression, including their interpretability, computational efficiency, and practicality in resource-limited or transparent decision-making scenarios. As noted in the manuscript, the choice between simple and complex models ultimately depends on the specific clinical application and the balance between performance gains and trade-offs in interpretability and computational demands.

We hope this expanded discussion adequately addresses your concern. Thank you again for your insightful feedback, which has helped us improve the clarity and depth of our manuscript.

Q6: Ethical implications of using the proposed automated decision making system in clinical should be briefly discussed.

A: Thank you for your thoughtful comment regarding the ethical implications of using the proposed automated decision-making system in clinical practice. This is indeed a critically important aspect of our study, and we have addressed it in the revised Discussion section (lines 412–424) .

While such systems can significantly enhance the accuracy and efficiency of triage processes, we emphasized that they should not replace human judgment but rather serve as supportive tools for healthcare professionals. We have also acknowledged potential risks, such as over-reliance on automated systems, which might overlook nuanced clinical contexts that are difficult to quantify or encode into models. Furthermore, we highlighted the importance of addressing algorithmic bias, which may stem from imbalanced datasets or unrepresentative training data, to ensure equitable care across patient populations.

In addition, we stressed that transparency in model development, validation, and implementation is essential for fostering trust among healthcare providers and patients. Most importantly, we underscored that the deployment of automated systems should always respect and prioritize patient autonomy, using predictions to inform, rather than dictate, clinical decisions.

We hope this addition addresses your concern and further strengthens the ethical considerations of our study.

Once again, we sincerely thank you for your thoughtful and constructive feedback. Your comments have been invaluable in improving the quality and clarity of our manuscript. While we have addressed the key points outlined in your review to the best of our current abilities, we recognize the significance of the remaining suggestions and will strive to incorporate them into future research where feasible.

We hope that our revisions and responses satisfactorily address your concerns, and we are happy to provide further clarifications if needed.

Thank you for your time and effort in reviewing our work.

Sincerely,

Wei Chong

Corresponding Author

Email: wchong@cmu.edu.cn

---

## [Decision Letter · Decision Letter 1]

7 Jan 2025

Development and validation of interpretable machine learning models for triage patients admitted to the intensive care unit

PONE-D-24-51129R1

Dear Dr. Liu,

We’re pleased to inform you that your manuscript has been judged scientifically suitable for publication and will be formally accepted for publication once it meets all outstanding technical requirements.

Kind regards,

Jerome Baudry, Ph.D.

Academic Editor

PLOS ONE

Additional Editor Comments (optional):

Reviewers' comments:

Reviewer's Responses to Questions

**Comments to the Author**

1. If the authors have adequately addressed your comments raised in a previous round of review and you feel that this manuscript is now acceptable for publication, you may indicate that here to bypass the “Comments to the Author” section, enter your conflict of interest statement in the “Confidential to Editor” section, and submit your "Accept" recommendation.

Reviewer #1: All comments have been addressed

2. Is the manuscript technically sound, and do the data support the conclusions?

Reviewer #1: Yes

3. Has the statistical analysis been performed appropriately and rigorously? 

Reviewer #1: Yes

4. Have the authors made all data underlying the findings in their manuscript fully available?

Reviewer #1: Yes

5. Is the manuscript presented in an intelligible fashion and written in standard English?

Reviewer #1: Yes

6. Review Comments to the Author

Reviewer #1: (No Response)

7. PLOS authors have the option to publish the peer review history of their article (what does this mean? ). If published, this will include your full peer review and any attached files.

**Do you want your identity to be public for this peer review?** For information about this choice, including consent withdrawal, please see our Privacy Policy .

Reviewer #1: **Yes: ** Armin Ahmadi

---

## [Editor Report · Acceptance letter]

PONE-D-24-51129R1

PLOS ONE

Dear Dr. Liu,

I'm pleased to inform you that your manuscript has been deemed suitable for publication in PLOS ONE. Congratulations! Your manuscript is now being handed over to our production team.

Kind regards,

on behalf of

Dr. Jerome Baudry

Academic Editor

PLOS ONE